Grytten *et al. Genome Biology*　　(2022) 23:209

**METHOD**

# KAGE: fast alignment-free graph-based genotyping of SNPs and short indels

Ivar Grytten[1,2]* , Knut Dagestad Rand[1,2] and Geir Kjetil Sandve[1,2]

*Correspondence:
ivargry@ifi.uio.no

[1] Department of Informatics, University of Oslo, Gaustadalleen 23 B, 0371 Oslo, Norway
[2] Centre for Bioinformatics, University of Oslo, Gaustadalleen 30, 0373 Oslo, Norway

**Abstract**

Genotyping is a core application of high-throughput sequencing. We present KAGE, a genotyper for SNPs and short indels that is inspired by recent developments within graph-based genome representations and alignment-free methods. KAGE uses a pan-genome representation of the population to efficiently and accurately predict genotypes. Two novel ideas improve both the speed and accuracy: a Bayesian model incorporates genotypes from thousands of individuals to improve prediction accuracy, and a computationally efficient method leverages correlation between variants. We show that the accuracy of KAGE is at par with the best existing alignment-free genotypers, while being an order of magnitude faster.

**Keywords:** Genotyping, Variant calling, Pangenomes, Graph genomes, Alignment-free

## Background

Recent studies such as the 1000 Genomes Project [1] have produced large catalogues of known human genetic variation. Researchers now have access to genotype information for thousands of individuals at known variant sites, providing unprecedented knowledge about how often genetic variants occur together at sub-population and individual levels.

Discovery and characterization of an individual's genetic variation have traditionally been done in two steps: 1) *variant calling*: discovering genetic variation in the genome, and 2) *genotyping* those called variants: determining whether each genetic variant is present in one or both of the chromosomes of the individual. Variant calling is a computationally expensive task, usually performed by first sequencing the genome of interest at high coverage, followed by mapping the sequenced reads to a reference genome and inferring variants according to where the mapped reads differ from the reference. It is now possible to skip the variant calling step, instead relying on catalogues of known genetic variation to directly genotype a given individual. For humans, the huge amount of already detected genomic variation means that a large amount (>90%) of an individual's genetic variation can be detected this way [1]. Genotyping individuals at sites of known variation has in principle been performed already since the early 2000's based on

SNP chips (microarrays). This has however been restricted to a limited set of fixed sites that a given chip has been designed to capture. While these chip-based techniques are still popular today due to their low cost, e.g. in GWAS studies [2], they are limited by chip architecture and thus only able to genotype about 1-2 million variants. Contrary to this, *sequence-based* genotyping techniques are, while being more expensive to perform, in theory able to genotype any genetic variant that has already been detected in the population.

Sequence-based genotyping is traditionally performed by aligning sequenced reads to a reference genome and examining how the reads support genotypes locally at each variant [3, 4]. While these methods generally have high accuracy, they are slow, mostly because read-mapping/alignment is computationally expensive. Also, since these methods require the mapping of reads to a reference genome, they have a tendency to perform poorly in regions where the reference genome is very different from the sequenced genome, such as in variant-rich regions. Furthermore, since reads are more likely to be correctly mapped when they are similar to the reference, reads supporting variant-alleles may be underrepresented among the mapped reads, resulting in a bias towards the reference alleles, a problem referred to as reference bias [5].

In an attempt to both speed up genotyping and avoid reference bias, several *alignment-free* approaches have emerged during the last few years [6–9]. These methods work by representing genetic variants by their surrounding kmers (sequences with length k covering each variant) and looking for support for these kmers in the sequenced reads. Since these methods do not map reads to a reference genome, they mitigate the problem of reference bias, and are usually computationally very efficient since kmer-lookup is fast compared to read mapping. However, these methods struggle when variant alleles cannot be represented by unique kmers, e.g. because a variant allele shares one or more kmers with another location in the genome. The genotyping method Malva [8] attempts to solve this problem by using larger kmers in cases where kmers are non-unique, but is still not able to genotype all variants using this approach. The more recent method PanGenie [10] takes the approach of Malva one step further and uses known population haplotype information to infer the likelihood of genotypes for variants that do not have unique kmers, such as variants in repetitive regions of the genome. PanGenie does this by using a Hidden Markov Model (HMM) with one state for each possible pair of haplotypes from the set of known haplotypes in the population. While this approach works well for relatively few input haplotypes and variants (PanGenie was tested using haplotypes from around 10 individuals and about 7 million variants), the number of states in the HMM increases quadratically with the number of haplotypes, and the method scales poorly when more than a few haplotypes are used. This means that the more than 5000 haplotypes and 80 million variants available in the 1000 Genomes Project cannot easily be utilised by an approach like PanGenie. Furthermore, PanGenie does not use kmer-information at all for variants with non-unique kmers.

We here describe KAGE – a new genotyper for SNPs and short indels that builds on recent ideas of alignment-free genotyping from Malva and PanGenie for computationally efficiency. KAGE implements two novel ideas for utilising all previously known haplotype information from repositories such as the 1000 Genomes Project in order to improve genotyping accuracy. We show that combining these ideas leads to a genotyper

**Step 1:** Index graph/variants with kmers

Input-vcf

| ID | Position | Ref | Alt |
|----|----------|-----|-----|
| variant1 | chr1:3 | G | T |
| variant2 | chr1:7 | A | ACG |

Graph

A C G A C T A T T

Kmer-index over variants in the graph (k=3)

| Variant ID | Ref-kmers | Alt-kmers |
|------------|-----------|-----------|
| variant1 | CTA | CGA |
| variant2 | ACG | ATT |

**Step 2**: Count kmers in reads

Reads

ACTACT
ACGACT
CGACTA
ACTATT

Kmer counts

CTA: 2
CGA: 2
ACT: 2
ATT: 1

**Step 3**: Compute genotype-probabilities

Using Bayes rule:

P(genotype|kmer counts) =
  P(kmer counts|genotype) * P(genotype in population) / ...

**Fig. 1** Overview of a typical alignment-free genotyping approach. First, kmers covering each allele of each variant is stored (step 1), e.g. by choosing kmers from a graph representation of the variants. All kmers from the reads are then collected (step 2), and genotype probabilities are computed using Bayes rule with genotype probabilities from a known population used as priors (step 3). The most likely genotype is chosen

that is both more accurate and computationally efficient than existing alignment-free genotypers.

## Results

We here present KAGE, which is based on two new ideas that are inspired by recent progress in alignment-free genotyping and imputation. We first show how each idea can improve genotyping accuracy for alignment-free genotyping methods, and then show that when these ideas are combined into KAGE, we get a graph-based alignment-free genotyper that is highly accurate and considerably more computationally efficient than existing alignment-free genotyping methods.

### Idea 1: Modelling expected kmer-counts from the population

Alignment-free genotypers, like Malva and PanGenie, compute genotype likelihoods by counting the number of kmers from the read set that support each allele of every variant to be genotyped, as illustrated in Fig. 1. A problem with this approach is that some variants may have alleles that are covered by kmers that also match elsewhere in the genome (for some or many individuals), making such variants difficult to genotype, as illustrated in Fig. 2. We hypothesise that this problem may be addressed by modelling the expected match count for each individual kmer based on population data, such as data from the 1000 Genomes Project.

To test this hypothesis, we implemented a baseline genotyper that follows the scheme in Fig. 1: each variant is represented by kmers, and the genotyper counts how many times each kmer exists in the read data sets, computing genotype probabilities using Bayes formula (naively assuming each kmer is unique in the genome). We also created a more sophisticated version of this genotyper, where we assume that each kmer exists with a given frequency in the population, and use this information when computing the probability of observing the given number of kmers for each variant (Section 5). We

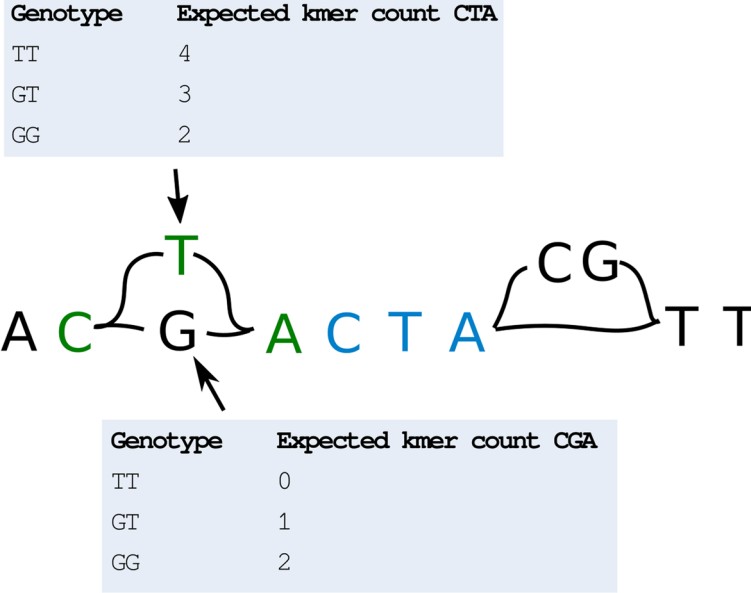

**Fig. 2** Graph with duplicate kmers. The SNP G/T has a kmer CTA (green) on the variant allele that also exists on the reference path (blue). If we observe each of the kmers CTA and CGA once in the read set, we might be fooled to believe that GT is the most likely genotype for this variant. However, when knowing that the kmer CTA is expected to occur at least once in the read data set, due to the duplication, we might conclude otherwise. This information can be used to adjust the probabilities used to compute the binomial probabilities of observing kmer counts given genotypes

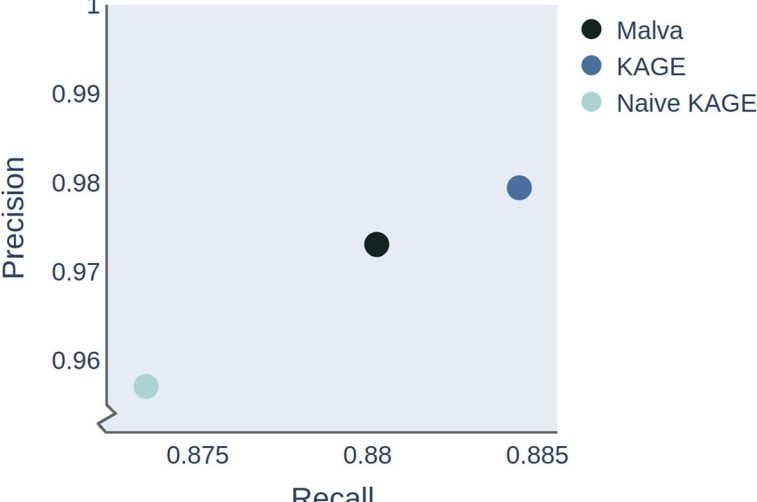

**Fig. 3** Genotype accuracy increases when expected kmer counts are modelled from the population. Our prototype alignment-free genotyper (naive KAGE) performs a bit worse than Malva, as expected. However, when we compute expected kmer counts from the population, using 1000 Genomes data, and use these counts to compute the probability of observing the given kmers in our data, the genotyping accuracy greatly improves. Note: KAGE here refers to a simplified version of the full genotyper that we present in the next section (here KAGE does not use variants to adjust prior probabilities)

benchmark both these methods against Malva. As shown in Fig. 3, our naive method performs worse than Malva, whereas the approach where expected kmer counts are modelled from the population performs better. In Table 1 we show the accuracy of the

two methods on SNPs that have unique and non-unique kmers. As can seen in the table, the accuracy on SNPs with non-unique kmers drastically increases for the more sophisticated genotyper that uses information about the kmer frequency from the population. Although this is an experiment performed on a single small benchmark dataset, these findings provide an indication that the idea of comparing observed kmer counts to expected kmer counts based on a population may work.

**Idea 2: Using a single variant to adjust the prior**

While Malva uses the allele frequency from the population as priors for genotype likelihoods, PanGenie uses a more sophisticated approach where genotypes are inferred using a Hidden Markov Model (HMM), meaning that the predicted genotype of one variant provides information that can support the determination of the genotypes of other variants. While this approach may allow PanGenie to predict genotypes with higher accuracy than Malva, this method does not scale to handle a large number of known haplotypes, since the HMM must represent every possible pair of haplotypes as a separate state, making the number of HMM states grow quadratically with the number of haplotypes. We hypothesise that a simpler approach may be better for two reasons: First, a simple approach may be able to utilise all the haplotype information available (e.g. the 2548 individuals available from the 1000 Genomes project, rather than only a handful that the PanGenie approach would typically employ), and second, such an approach may be computationally faster than an HMM-approach that requires inference on hidden states. To test this hypothesis, we implement a way of efficiently computing prior probabilities for each genotype of each variant given genotype probabilities of other variants (Section 5). We let every variant only depend on a single other variant, which leads to a very efficient way of genotyping all variants, and allows us to use as many underlying haplotypes as we would like. The determination of a single best suited helper variant for each known variant in the genome is performed only once, in a general pre-processing step. The genotyping of individuals then only needs to perform a fast lookup in this precomputed table. We test this approach against PanGenie, where PanGenie is run with a varying number of known haplotypes as input. In order to test PanGenie with as many haplotypes as possible from the 1000 Genomes Projects, we perform this experiment on a small test data set (5 million base pairs of chromosome 1).

The result of this experiment is shown in Fig. 4, where it can be seen that the accuracy of both this approach (KAGE) and PanGenie increase with the number of known haplotypes. This experiment is performed on a small test dataset, so as to allow the inclusion of a larger number of individuals in PanGenie, and is not meant as a benchmarking of the methods per se. These findings provide an indication that the use of a single variant as prior still allows us to leverage information from a large number of individuals.

**A new graph-based alignment-free genotyper**

We combine the two ideas discussed above into a new alignment-free genotyper KAGE, and compare this genotyper in terms of running time, memory usage and accuracy against existing alignment-free genotypers as well as the most commonly used alignment-based genotypers. We follow the guidelines from "Best practises for benchmarking

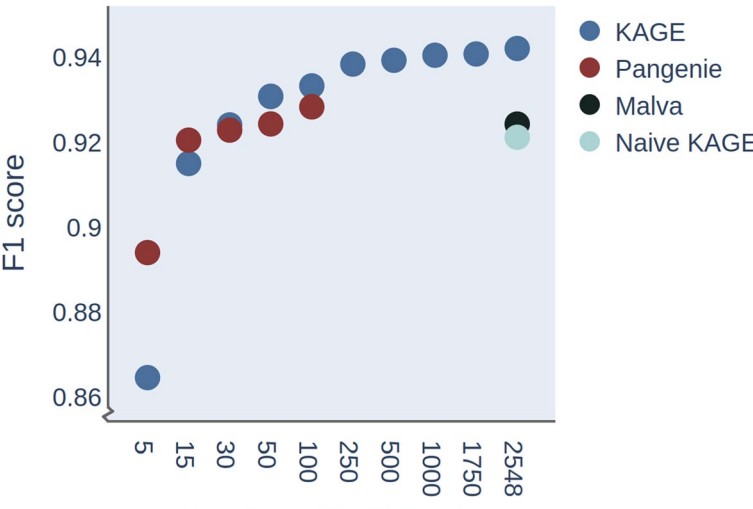

**Fig. 4** Accuracy of Pangenie and KAGE as a function of the number of individuals included in the model on a small test dataset. Recall and precision (shown as F1 score) are computed for the predicted genotypes by the various methods on a small test dataset (genome size 5 million bp) with reads simulated from HG002, in order to test whether the accuracy of PanGenie and KAGE increases with a larger number of individuals in the model. The F1 score of Malva and "our naive approach" are shown as a single dot, since both of these only use the whole population genotype frequency when estimating genotype probabilities

germline small-variant calls in human genomes" [11] and compare the output of each genotyper against a truth data set (similarly to how PanGenie and Malva assess their methods).

In the following experiments, we run PanGenie with 64 input haplotypes (from 32 individuals), since a larger number of haplotypes make PanGenie too slow and memory demanding when run on whole-genome data. We run KAGE with all of the 5096 haplotypes available from the 1000 Genomes Project. When listing run times, we consider only the time used to genotype a given sample, not the time used to index the reference genomes, graphs, or variants, since such indices can be created once and then be re-used for genotyping an arbitrary number of samples. All experiments are based on genotyping variants from the 1000 Genomes Project using short reads as input. Accuracy is presented by recall and precision when predicted genotypes are compared against a truth dataset. Further details about these experiments can be found in the Section 5, and a Snakemake pipeline for reproducing all the results can be found at https://github.com/ivargr/genotyping-benchmarking.

In the experiments, we compare the genotypers listed in Table 2, which includes the alignment-based genotyper Graphtyper as well as all alignment-free genotypers we believe are relevant or commonly used. For comparison, we also run the commonly used variant caller GATK [3], which does not only genotype a specified set of variants, but tries to detect any variation from the reference. We include GATK so that the results of the genotypers can be contrasted to the alternative approach of performing full variant calling.

We genotype the individual HG002 from GIAB, which is an individual commonly used to benchmark variant callers and genotypers. The results can be seen in Table 3 and Fig. 5. We ran the final version of KAGE also on another individual (HG006, also

from GIAB), which was not consulted until after we finalised all method and parameter choices, and verified that the results are similar (Additional file 1: Table S2 and S3). The results from genotyping HG002 using 30x read coverage can be found in Additional file 1: Table S1, and are similar to the results from 15x coverage. The experiments performed show that KAGE is about as accurate as, or more accurate than, all the other genotypers and considerably faster than any of the methods. GATK comes out as clearly more accurate than the genotype methods, but is as expected considerably slower than most of the genotypers. KAGE is about as accurate as the alignment-based method Graphtyper while being more than 30 times faster. KAGE is able to genotype a sample fast by pre-building all indexes (Section 5). While creating these indexes is quite time-consuming (a couple of days), once indexes are built for a set of variants and a reference population, genotyping a new sample takes a short fixed amount of time, which is independent of the number of individuals in the reference population. For the experiment presented in Table 3, KAGE spends about 2 minutes on genotyping and 10 minutes on counting kmers.

The idea of using a single variant to adjust the prior probabilities when genotyping can be seen as a way of using information at some variants to "impute" the genotypes at other variants. While this seems to work well for KAGE, and leads to fast genotyping, we were curious as to whether higher accuracy could be achieved by using established and more sophisticated imputation tools. To test this hypothesis, we ran KAGE without the builtin imputation, producing a VCF with genotype likelihoods that reflect the probability of the possible genotypes given only kmer information. We then ran the imputation tool GLIMPSE [12] on this VCF. As seen in Table 3 (KAGE + GLIMPSE), this combination of KAGE and GLIMPSE leads to higher accuracy than only running KAGE, showing the strength of imputation to improve genotyping accuracy.

## Discussion

We find that KAGE has as high or higher accuracy than the other genotypers. PanGenie seems to perform similarly on indels, but performs slightly worse on SNPs. Malva performs worse than all the other methods, which we believe makes sense considering that Malva does not have a good way of dealing with variants represented by non-unique kmers. While PanGenie has almost as good accuracy as KAGE, it requires more than 20 times the run time and considerably more memory than KAGE on the experiments we have run. KAGE is able to genotype a full sample with 15x coverage in only about 12 minutes using 16 compute cores, while all the other methods require several hours. This means that with KAGE, given pre-built indexes, it is now possible to genotype a sample quickly and easily on a standard laptop, such as in a clinical setting. The computational efficiency of KAGE also allows for a reduced energy footprint when large numbers of samples are to be genotyped. This is highly relevant given current plans of sequencing more than a million genomes in the coming years [13].

We confirm that performing full variant calling with a method such as GATK yields considerably higher accuracy than only genotyping a preset of specified variants. One should therefore be aware of this speed/accuracy tradeoff: genotyping a preset of known variants can be very efficient and give decent accuracy now that a lot of genomic variation is known. However, when the aim is to accurately discover as many variants in

a sample as possible, and runtime is not an issue, a variant caller like GATK or Deep-Variant [14] would be the best option. We note that GATK has higher accuracy in our experiments than what has been found in the previous benchmarks presented in the Graphtyper [4] and Malva [8] papers, where Malva and Graphtyper are both shown to have as good as and in some cases better genotyping accuracy than GATK. We believe this might be because GATK has been significantly improved in the last few years since these papers were published [3]. It should be noted that GATK was used among other tools to create the GIAB truth dataset, which could potentially influence the measured performance of GATK in our experiments.

We are surprised to see that KAGE and PanGenie, which are completely alignment-free, are able to achieve very close accuracy to Graphtyper, which first maps and aligns all reads using BWA-MEM and then locally realigns all reads to a sequence graph. We believe the reason must be that KAGE and PanGenie exploit population information in a more sophisticated way when computing the genotype likelihoods. We speculate that PanGenie should be able to achieve even better accuracy by using more known haplotypes as input, but this would lead to very long run-time and high memory usage due to the quadratic complexity of the Hidden Markov Model used.

While KAGE's fairly simple builtin "imputation" that uses only a single variant to help genotype another variant seems to work well, we observe higher accuracy when running the more sophisticated imputation tool GLIMPSE on the output from KAGE. We have developed KAGE so that it can optionally be run without the builtin imputation, so that any other imputation tool can be run on the output if the user wishes. We believe that making software modular like this is beneficial for the bioinformatics community, as it enables coupling of different software that are specialised in certain tasks. We believe the high accuracy achieved by KAGE and GLIMPSE together highlights the strength of using reference populations/pangenomes to improve genotyping performance, since in our experiments, this combination clearly outperforms all the other genotypers in run-time, and all except GATK in accuracy.

While Malva's strength is supposed to be a relatively low run-time, Malva is unfortunately not optimised to run on multiple compute cores/threads. We believe Malva would be able to run considerably faster if it was able to utilise multiple compute cores in the genotyping step. We observe that Bayestyper differs from the other methods by having a lower recall and a higher precision, meaning that Bayestyper is likely more conservative when genotyping. We were not able to tune Bayestyper to provide higher recall at the cost of lower precision.

There are a few limitations of KAGE. First, KAGE is for now only able to genotype SNPs and short indels, and it is important to note that our benchmarks only compare the tools on SNPs and short indels, while several of the methods have been developed also for structural variation. We believe, however, that the ideas that KAGE are based on can be generalised to structural variant calling with short reads, which would be an interesting opportunity for further work. Second, KAGE relies on a relatively good database of known variation from a population. This is important to remember, as not all individuals are well represented in e.g. the 1000 Genomes Project, and genotyping accuracy for such individuals is likely to be lower. The fact that more and more genotypers are using population information to improve accuracy highlights the

importance of good reference populations (pangenomes) that represent and cover all ancestries. In the same way, KAGE will not work on other species where pangenomes are not yet available. This does, however, also mean that we expect KAGE to potentially perform even better in the future when more genetic variation data from more individuals will be available. Finally, KAGE and the other kmer-based methods are not able to use information from paired-end reads. We do not consider this a major problem, however, since single-end sequencing is very common and cheaper to perform than paired-end sequencing.

## Conclusions

KAGE allows genotyping of SNPs and short indels by an alignment-free approach that is more than 20 times faster than existing methods, while offering competitive accuracy. We see KAGE as a highly timely contribution to the field, given the plans of large international consortia to sequence and genotype millions of genomes in the coming years.

## Methods

### Model description

The following describes the model and assumptions used in KAGE. We here assume all variants are biallelic variants, each with two *alleles* – a *reference allele* and an *alternative allele*. An individual can thus have the following three genotypes at a variant: Having the reference allele in both chromosomes (we refer to this as having the genotype 0/0), having the alternative allele in both chromosomes (1/1), or having the reference allele in one chromosome and the alternative allele in the other (0/1). Note that we are not concerned with the phasing of genotypes, so 1/0 and 0/1 are considered as being the same genotype.

Assuming a variant of interest $i$, we define $P(G_i = g)$ as the probability of having the genotype $g$ at this variant, where $g$ is either 0/0, 0/1 or 1/1. We let $K_i = (K_{ir}, K_{ia})$ be the observed kmer counts on the reference and variant allele of variant $i$. Every variant of interest $i$ has a preselected helper variant $h$ with kmer counts $K_h$. KAGE genotypes a variant $i$ by selecting the most likely genotype. This is done by calculating the posterior probabilities $P(G_i|K_i, K_h)$:

$$P(G_i|K_i, K_h) = P(K_i|G_i) \sum_{G_h \in G} P(G_h)P(G_i|G_h)P(K_h|G_h) \tag{1}$$

In the above formula, $P(K_i|G_i)$ is the likelihood of observing the kmer counts $K_i$ on the variant of interest given a genotype $G_i$. The summation corresponds to a prior probability of the genotype $G_i$ given the observed counts on the helper variant and the population structure, with the individual probabilities corresponding to the following:

- $P(G_h)$ is the probability (over the population) of genotype $G_h$ at the helper variant.
- $P(G_i|G_h)$ is the probability (over the population) of having genotype $G_i$ on the variant of interest given genotype $G_h$ at the helper variant

- $P(K_h|G_h)$ is the likelihood of observing the kmer counts $K_h$ given genotype $G_h$ at the helper variant, similarly to $P(K_i|G_i)$.

The probability distributions for $P(K_i|G_i)$ and $P(K_h|G_h)$ are modelled as mixtures of Poisson distributions, one Poisson distribution for each individual in the reference population that has the given genotype:

$$P(K_{ir}|G_i) = \sum_{t:g_i(t)=G_i} Pois(K_{ir}; \lambda_{irt})/|\{t : g_i(t) = G_i\}|$$

where $g_i(t)$ is the genotype of individual $t$ at variant $i$. The rate $\lambda_{irt}$ of each individual is proportional to the number of times the kmer $s_{ir}$ occurs in that individual's genome ($d_{irt}$) (plus an error term). This gives us:

$$P(K_{ir}|G_i) = \sum_{t=1:g_i(t)=G_i}^{n} Pois(K_{ir}; \lambda_0(d_{irt} + \epsilon))/|\{t : g_i(t) = G_i\}|$$

We note that, implementation-wise, the above sum is calculated in a more efficient manner.

The corresponding probabilities $P(K_{ia}|G_i)$, $P(K_{hr}|G_r)$ and $P(K_{ha}|G_a)$ are calculated using the same setup. The next sections describe further how this is implemented in KAGE, and we refer to the Additional material for more details.

### KAGE implementation

Given the model described in the previous section, the following explains more in detail how KAGE works. KAGE roughly consists of three steps: *indexing*, *kmer counting* and *genotyping*.

### *Indexing: Creating a kmer-to-graph-node index, choosing helper variants and finding expected kmer counts*

All the following indexes are built once for a reference population and a set of variants one wants to genotype.

In order to genotype a variant, KAGE needs to know how many kmers in the read data set support each allele of the variant. For this, we build an index that enables lookup from any kmer to nodes (if any) in the graph the kmer maps to. Given a set of bi-allelic variants, kmers representing each allele are selected so that every allele of every variant is represented by at least one kmer. This is done by representing the variants as a genome graph (using the Python package obgraph), and for each variant considering all kmers covering the alleles of the variants. A set of kmers is chosen by trying to pick the kmers with lowest maximum "population frequency", where population frequency is the expected number of times this kmer is found in the population (Fig. 6). The resulting kmers are stored in an index that enables direct lookup from any kmer to the nodes in the graph that the kmer covers.

For each variant of interest, we also want to find one other variant that correlates well with our variant of interest. This means that for a good helper variant, individuals tend

to have the same genotype on the helper variant as on the variant of interest. Using such variants we can improve our prediction of the interest-genotype in cases where the count model for the variant of interest gives ambiguous results. In order to find good helper variants, we evaluate the metric $\sum_{G \in \mathcal{G}} \log \pi_\nu(G, G)$ for the neighbouring 200 variants to the variant of interest, and choose as helper the variant $\nu$ with the highest score.

KAGE also needs a model for expected kmer counts for a "random" individual in the population. For each possible genotype at a variant, we simply count, for each variant allele, how many individuals with the given genotype in a reference population have 0, 1, 2, 3, $\cdots$ and so on kmer counts on that allele and store this information in an index.

### Counting kmers in a read data set

Given the kmer indexes described in the previous section, we can count how many times kmers from a read data set maps to nodes representing variant alleles in the graph. The result from this procedure is a count for each node in the graph, saying how many times kmers from the read data set mapped to that node.

### Genotyping

KAGE genotypes a bi-allelic variant in the following way. Given kmer counts for each allele of the variant, the probabilities of observing those counts given the different possible genotypes are calculated using combinations of Poisson models. These models take into account how common it is to observe various kmer counts in a reference population. Similar probabilities are also calculated for the helper variant. We can then calculate the probability of getting the observed counts given each combination of genotypes on the two variants (9 combinations). When combining this with the prior probabilities of each genotype combination (calculated from the observed genotype-combinations in a reference population), we can use Bayes rule to calculate the probabilities of the different genotype combinations given the observed counts. In order to find the marginal probabilities for each genotype of the main variant, we simply sum over the probabilities for each genotype on the helper variant. A more mathematical description is given previously under the section *Model description*, and the full details can be found in the additional material.

### Benchmarking genotypers

We compared KAGE against the following other tools: GATK HaplotypeCaller version 4.2.3.0, Graphtyper version 2.7.1 (using the genotype subcommand), Malva version 1.3.1, PanGenie (commit ID da87f55cdccf31cd2c0008fd4848e33ba42021fc) and Bayestyper version 1.5. We also tried including Platypus in the comparison, but were not able to get the tool to work. Both Bayestyper and Malva use kmer-counts from kmc [15]. We ran kmc version 3.1.1 and included singleton kmers (using the flag -ci1) for Bayestyper and Malva, as this seemed to give the best results. For Bayestyper and Malva, we used the suggested k specified by the tool documentation, which was 51 for Bayestyper and 31 (short kmers) and 45 (long kmers) for Malva. We ran KAGE and PanGenie with the recommended k=31. While GATK has the option to genotype a specified set of variants, we did not use this option. The reason is that even when this option is specified,

**Table 1** Accuracy on SNPs that can be represented with unique kmers vs. SNPs that cannot. The accuracy increases for variants that cannot be represented with unique kmers after implementing idea 1 (KAGE with model of kmer counts) and increases further when introducing idea 2 (KAGE full). The accuracy on SNPs that can be represented with unique kmers is always high, since these SNPs are easy to genotype

| Method | Accuracy on SNPs with non-unique kmers | Accuracy on SNPs with unique kmers |
|---|---|---|
| Naive KAGE | 64 % | 97 % |
| KAGE with model of kmer counts | 77 % | 97 % |
| KAGE (full) | 92 % | 99 % |

**Table 2** Overview of genotypers

| Genotyper | Alignment-free/alignment-based | Uses prior haplotype-information |
|---|---|---|
| Malva | Alignment-free | No |
| Bayestyper | Alignment-free | No |
| PanGenie | Alignment-free | Yes |
| Graphtyper | Alignment-based | No |
| GATK | Alignment-based | No |

**Table 3** Results from genotyping HG002

| | Indels recall | Indels precision | Indels F1 | SNPs recall | SNPs precision | SNPs F1 | Runtime | Memory usage |
|---|---|---|---|---|---|---|---|---|
| KAGE | 0.582 | 0.908 | 0.709 | 0.929 | 0.981 | 0.955 | 12 min | 18 GB |
| KAGE + GLIMPSE | 0.594 | 0.894 | 0.713 | 0.948 | 0.99 | 0.968 | 1.4 hours | 18 GB |
| PanGenie | 0.561 | 0.926 | 0.699 | 0.905 | 0.979 | 0.941 | 3.9 hours | 111 GB |
| Bayestyper | 0.485 | 0.993 | 0.652 | 0.834 | 0.997 | 0.908 | 7.9 hours | 30 GB |
| Malva | 0.537 | 0.856 | 0.659 | 0.849 | 0.921 | 0.883 | 14.9 hours | 50 GB |
| Graphtyper | 0.568 | 0.941 | 0.708 | 0.909 | 0.993 | 0.949 | 5.1 hours | 23 GB |
| GATK | 0.898 | 0.96 | 0.928 | 0.966 | 0.981 | 0.973 | 9.0 hours | 64 GB |

GATK will also call other alleles at the specified sites. We also experienced slower run time using this option. Thus, we chose to let GATK just do full variant calling, which is also what it is built for.

GATK and Graphtyper were run on reads mapped to GRCh38 using BWA-MEM [16] version 0.7.17-r1188. A Snakemake [17] pipeline for running all the experiments and a conda environment with all the necessary tools can be found at https://github.com/ivargr/genotyping-benchmarking. Reads were simulated using Graph Read Simulator (version 0.0.9) with substitution, deletion and insertion rate all set to 0.001. When using experimental data all reads were obtained from the GIAB repository and downsampled to the given coverage used in the experiment. Exact data URLs can be found in the config file of the Snakemake pipeline.

Variants genotyped in the experiments are from the 1000 Genomes Project v2a. Only variants with allele frequency >0.1% were used. Accuracy (precision and recall)

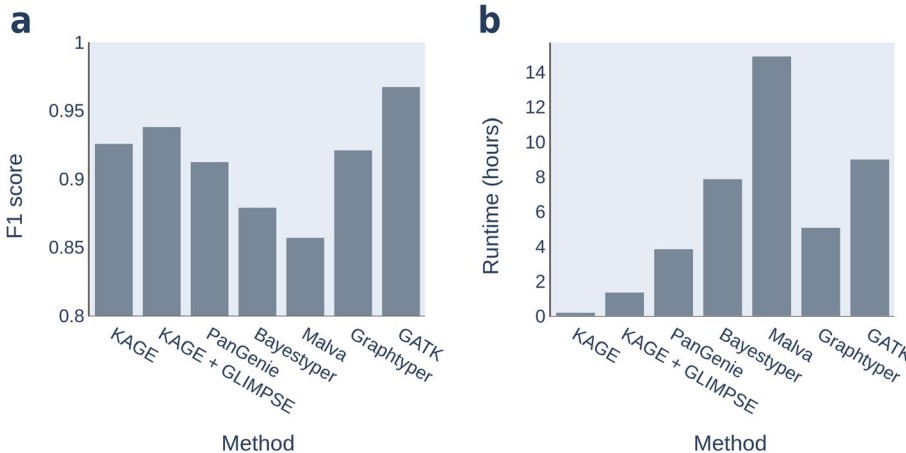

**Fig. 5** Accuracy and runtime of methods. Showing **a** the F1 score on SNPs and Indels combined and **b** the runtime of each of the methods when genotyping HG002 using read data with 15x coverage. All tools, except Malva, are run with 16 compute cores

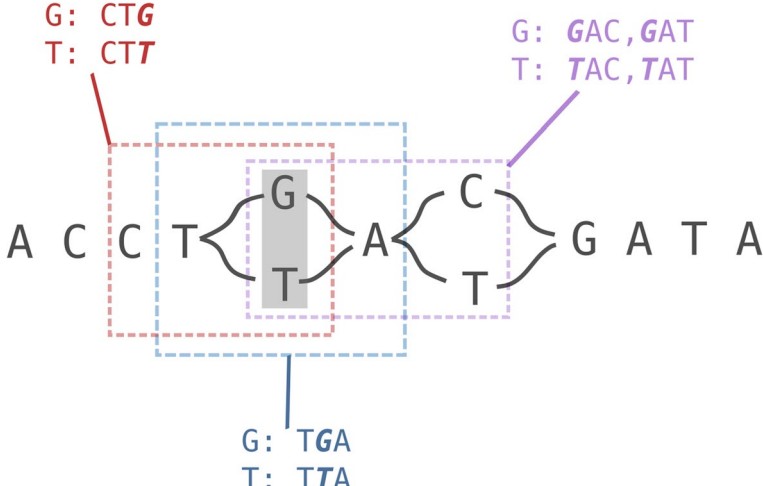

**Fig. 6** Illustration of kmer selection. Example of how 3-mers can be selected for the SNP T/G (grey box). KAGE uses a graph-representation of all variants, and considers all possible ways to pick kmers around the two alleles of a variant. Here, there are three possible ways to pick a set of kmers, illustrated by red, blue and purple boxes. In this case, the kmers marked in red will be preferred, as the blue kmers include TTA which is not unique (exists elsewhere in the graph), and the purple kmers similarly include the non-unique kmer GAT

was measured using Happy version 0.3.14 against GIAB truth datasets [18]. Accuracy was measured within high-confidence regions using the Happy -f flag with the regions file that accompany each GIAB data set. We did not filter any variants on quality or other criteria before running Happy.

To create Table 1, we for simplicity only measure the accuracy on SNPs that exist in the truth dataset, and define a correctly genotyped variant as a variant with the same predicted genotype as the genotype of the variant in the truth data set. The percentages are computed by dividing by the number of correctly genotyped variants with the total number of variants in the truth data set that have unique/non-unique kmers. A variant is defined as having unique kmers if, for both alleles of the variant, there

exists at least one kmer that does not exist anywhere else in the genome graph that represents all the variants in the population.

### Software implementation

KAGE was implemented using Python 3, and some run-time critical code (kmer lookup) were written in Cython. All code developed as part of this method has been modularized into four Python packages, each available through the PyPi package repository: kage-genotyper (main tool for genotyping), graph_kmer_index (building and using the kmer indexes necessary for the genotyping), kmer-mapper (counting kmers) and obgraph (building and working with the sequence graphs). KAGE with all dependencies can be installed directly by installing only the kage-genotyper package.

## Supplementary Information

> Additional file 1. Includes supplementary Tables S1-S3 and further details of method.
>
> Additional file 2. Review history.

**Peer review information**

**Review history**
The review history is available as Additional file 2.

**Authors' contributions**
IG, KDR and GKS drafted the manuscript. IG and KDR developed the software. IG, KDR and GKS contributed to the conception and design of the study, and to the interpretation of data. All authors have read and approved the final version of the manuscript.

**Funding**
This work was supported by the Centre for Computational Inference in Evolutionary Life Science (CELS). We also acknowledge generous support by the Research Council of Norway for an IKTPLUSS project (#311341) to KR and GKS.

**Availability of data and materials**
The datasets generated and/or analysed during the current study are available through the following git repository: https://github.com/ivargr/genotyping-benchmarking (version 0.0.2). An index that can be used with KAGE to reproduce the experiments is available at https://zenodo.org/record/6674055/files/index_2548all.npz. KAGE is available at https://github.com/ivargr/kage (version 0.1.1, GNU General Public Licence v3.0) and through Zenodo (DOI: 10.5281/zenodo.7065145).

## Declarations

**Ethics approval and consent to participate**
Not applicable

**Consent for publication**
Not applicable

**Competing interests**
The authors declare that they have no competing interests.

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

## 
