## [Additional file 2. Review history. · Genome Biology]

Review History

First round of review

Reviewer 1

Were you able to assess all statistics in the manuscript, including the appropriateness of statistical tests used? Yes: The employed statistical methods are appropriate.

Were you able to directly test the methods? Yes.

Comments to author:

The authors present a simple approach that allows for the alignment-free genotyping of variants in a variant call report (VCF file). The method has some limitations: it is only applicable to small variants (at least the authors only test these) and it only works on diploid genomes. It also does not use population information in the inference process. However, the authors indicate that these apparent drawbacks actually have little effect on the overall accuracy of the method, when compared to similar approaches like PanGenie. This is a somewhat surprising, but also encouraging result, because although it promises to reduce computational costs relative to alignment-based methods, and scale to large populations, it is frankly very difficult to use PanGenie to do so. In fact (the authors may want to point out), PanGenie does not use full population information during inference, due to quadratic/exponential costs in the number of considered haplotypes, it must select a subset of no more than 30 haplotypes for phasing, and for genotyping it clusters the population into no more than 30 bins. These limitations apparently render the benefits of working with the haplotype panel moot relative to the authors' approach.

I find the manuscript simple and compelling. I commend the authors on focusing on an immediate and practical solution. However, it seems to me that more work needs to be done to make KAGE something that can be used in production. In particular, the question of what kind of variants can be used remains unresolved. Can we attempt to genotype structural variants? At what scale does the method break down? Due to the great interest in pangenomics in crop breeding research, there is presently great demand for a pangenomic genotyping method that works on polyploids. It would seem that the authors could use math for polyploid genotyping such as that described in freebayes or other methods. Handling either or both of these features would help to set KAGE apart from the competition not only in runtime and accuracy, but also in feature set. I also wonder if the authors have considered using extremely long kmers to genotype larger complex variants.

It is a weak criticism, because it lies outside of what I think should be requested of the authors, but the use of python here seems problematic for the future maintenance and distribution of the software. A self-contained application in a lower-level language would potentially provide a much longer life to this software. In my experience, it can be extremely difficult to install old python packages. A system that can be built without reliance on external package managers might be advantageous in the long term.

Reviewer 2

Were you able to assess all statistics in the manuscript, including the appropriateness of statistical tests used? Yes: appropriate.

Were you able to directly test the methods? Yes.

Comments to author:

The article presents KAGE, a speedy alternative to alignment-free genotyping SNPs and short indels using non-paired-end short reads, with performance comparable to previous kmer-based methods. In particular, the authors propose to use evidence at one other "helper" site to aid in genotyping a given site.

Overall, my enthusiasm is reduced by the vagueness of the manuscript (especially of the Methods) and the lack of comparison against imputation-based tools like GLIMPSE.

Major comments:

The paper needs to clarify how "alignment free" methods are (or are not) related to avoiding reference bias. For instance, it is very possible for an alignment-free method to have reference bias. E.g. consider a k-mer-based genotyper that visits each polymorphic site in the genome and extracts two versions of the overlapping k-mers, one with the REF and one with ALT version of the site. If the flanking k-mer sequence consists of all REF alleles, then reference bias is still present. The paper should do more to explain that "alignment" is not the reason for reference bias; rather, the strict use of a single linear reference (whether we're doing alignment or genotyping) is the cause.

The major methodological idea is the use of evidence at a "helper" cite for genotyping. This takes a step outside the space of pure genotyping and toward something a little more like genotype imputation. Put another way: Li & Stephens based imputation tools (e.g. Beagle 5, GLIMPSE) are like KAGE but allow for a larger collection of nearby sites to influence the genotype at a site, via the HMM. When we think of it this way, it becomes important to also understand how KAGE compares to tools that specialize in imputation based on sequencing data, like GLIMPSE.

The description of the KAGE methods is extremely short; only about a page or less of the Methods section is actually dedicated to KAGE. We're left to guess many details.

The sense in which KAGE uses a "graph" was not well explained. The word "graph" is not mentioned in the Methods, except to say that a couple of the software components use graphs. Is KAGE graph-based only in the (rather trivial) sense that it considers both alleles at polymorphic sites? Where does the "graph" shape enter into the algorithm?

For speed comparisons, it seems that KAGE is being run on many cores at least some of the other tools are not. There should be at least one comparison where the tools use an equal number of cores. It's fine to also argue that KAGE scales well/better to many cores, but the speed comparison should be apples-to-apples in terms of what resources are actually made available to the algorithm. Also, the number of cores available to the algorithms should be laid out in the caption of any tables or figures presenting speed results.

More information on how the GIAB truth datasets are used to evaluate accuracy stats (data links, filtering procedures when plugging Hap.py, inclusion of high-confidence regions/bed files for different samples) would be helpful for quick reference.

Minor comments:

"While PanGenie has decent accuracy" -- replace "decent" with something specific.

PanGenie was sometimes misspelled.

The text should express some idea of how time and memory usage break down across the different components of KAGE, like 'graph-kmer-index', 'obgraph', and 'kmer-mapping'. I am also curious how these scale with the number of alleles being considered in the "panel."

Reviewer 3

Were you able to assess all statistics in the manuscript, including the appropriateness of statistical tests used? Yes: All stats are reasonable.

Were you able to directly test the methods? No.

Comments to author:

Grytten et al present a new method for alignment-free variant genotyping that performs as well as other methods but much faster and with lower memory. Overall the paper is well written and the results are compelling, but the specific use case is not well motivated. The authors do say this can run on a laptop "such as in a clinical setting", but I cannot imagine how this would happen given the indexing requirements. I do not mean to imply that KAGE has no use, I would just like the authors to better justify its need.

I do not understand the 1KG reference ([1]) in the background. The authors are discussing variant detection and give a 90% figure. The closest thing I can find in [1] is an imputation result, which is not variant detection. The authors could certainly derive this number in their experiments, and I suggest they do those experiments to give readers a more complete picture. Regardless of the validity of the reference, the 90% cannot be given without context. What allele frequency range is this 90% valid for and how does the % change for common and rare variants? The % is almost certainly ancestry specific. Genotyping individuals from populations with poor sampling will perform worse. How much worse is the recovery for different populations? What regions of the genome are considered in this %? Several papers over the previous few years have many many mega bases of new sequences in non-European samples. I suggest the authors incorporate these complexities (allele freq, ancestry, genomic region) into their larger analysis. Ancestry in particular must be considered. The negative effect of poor sampling for marginalized populations is well known in population genetics. We must work to improve sampling (which is clearly out of scope here) while also quantifying the effect on all methods going forward.

The authors hypothesize that performance is based on kmer uniqueness. I suggest they test this hypothesis. They can, for example, compare their naive and KAGE methods and show that outliers are correlated with repeats.

In another section the authors speculate about why KAGE and PanGenie do well but do not test this speculation. This speculation should be tested. They can, for example, start with a small amount of population information (eg just a few haplotypes for PanGenie) then scale up to thea

current result. This would not be too computationally expensive but would provide evidence in support of the speculation.

Was GATK used to define the truthset? If so that would explain GATK's performance and should be pointed out to the readers.

I was not able to test this software because "kage test" gave an error.

```
~$ kage test
```

```
Traceback (most recent call last):
```

```
File "HOME/.pyenv/versions/3.9.1/bin/kage", line 33, in <module>
```

```
sys.exit(load_entry_point('kage-genotyper==0.0.21', 'console_scripts', 'kage')())
```

```
File "HOME/.pyenv/versions/3.9.1/lib/python3.9/site-
```

```
packages/kage/command_line_interface.py", line 45, in main
```

```
run_argument_parser(sys.argv[1:])
```

```
File "HOME/.pyenv/versions/3.9.1/lib/python3.9/site-
```

```
packages/kage/command_line_interface.py", line 681, in run_argument_parser
```

```
args.func(args)
```

```
File "HOME/.pyenv/versions/3.9.1/lib/python3.9/site-
```

```
packages/kage/command_line_interface.py", line 330, in run_tests
```

```
run_genotyper_on_simulated_data(genotyper, args.n_variants, args.n_individuals,
```

```
args.average_coverage, args.coverage_std, args.duplication_rate)
```

```
File "HOME/.pyenv/versions/3.9.1/lib/python3.9/site-packages/kage/simulation.py", line 24, in
```

```
run_genotyper_on_simulated_data
```

```
genotypes, probs, _ = g.genotype()
```

```
File "HOME/.pyenv/versions/3.9.1/lib/python3.9/site-
```

```
packages/kage/combination_model_genotyper.py", line 156, in genotype
```

```
self.predict()
```

```
File "HOME/.pyenv/versions/3.9.1/lib/python3.9/site-
```

```
packages/kage/combination_model_genotyper.py", line 120, in predict
```

```
node_count_model_name = to_shared_memory(node_count_model)
```

```
File "HOME/.pyenv/versions/3.9.1/lib/python3.9/site-
```

```
packages/shared_memory_wrapper/shared_memory.py", line 323, in to_shared_memory
```

```
array_to_shared_memory(shared_memory_name, data, "python" if use_python_backend else
```

```
"shared_array")
```

```
File "HOME/.pyenv/versions/3.9.1/lib/python3.9/site-
```

```
packages/shared_memory_wrapper/shared_memory.py", line 104, in array_to_shared_memory
```

```
sa.delete(name)
```

```
OSError: [Errno 63] File name too long: '1288721527800148__frequencies_squared'
```

We thank all the reviewers for very useful feedback. Below we provide point-by-point-answers (in blue) to all the comments and questions. We have also done some minor changes to the manuscript and to KAGE that are not directly related to the reviewer comments, and for clarity we list these changes here first:

- 1) We have done some changes to how KAGE models expected kmer counts from the population. Thus, some parts of the methods section and supplementary material has changed to reflect the new model. All changes are marked in orange colour in the manuscript. Some of the results figures, especially those where smaller data sets and fewer individuals have been used, look slightly different, and some of the results for KAGE are now better, but no overall conclusions have changed.
- 2) We have been able to make Malva perform better on the small data sets by changing some parameters on kmc (changing minimum kmer frequency from 2 to 1 [the Malva documentation does not list any recommended parameters]). Thus Figure 1 and figure 2 have changed slightly (Malva performs better), but the conclusions still remain the same.
- 3) In figure 3, where we show that a genotyper that models kmer counts from a population performs better than MALVA, we initially compared “Naive KAGE” and Malva to the full KAGE genotyper. However, we think this figure makes more sense if we compare to a version of KAGE that does not use the concept of helper variants/imputation, since the point of the figure is only to show the effect of modelling kmer counts, and the notion of using helper variants is introduced later, in the next section.

Reviewer #1

The authors present a simple approach that allows for the alignment-free genotyping of variants in a variant call report (VCF file). The method has some limitations: it is only applicable to small variants (at least the authors only test these) and it only works on diploid genomes. It also does not use population information in the inference process. However, the authors indicate that these apparent drawbacks actually have little effect on the overall accuracy of the method, when compared to similar approaches like PanGenie. This is a somewhat surprising, but also encouraging result, because although it promises to reduce computational costs relative to alignment-based methods, and scale to large populations, it is frankly very difficult to use PanGenie to do so. In fact (the authors may want to point out), PanGenie does not use full population information during inference, due to quadratic/exponential costs in the number of considered haplotypes, it must select a subset of no more than 30 haplotypes for phasing, and for genotyping it clusters the population into no more than 30 bins. These limitations apparently render the benefits of working with the haplotype panel moot relative to the authors' approach.

We fully agree on the limitations pointed out by the reviewer: KAGE only works on diploid genomes and the manuscript only focuses on SNPs and short indels.

Since the initial submission, we have continued developing KAGE, and we have generalised the methodology so that KAGE now in practice can genotype structural variation. However, we have chosen to still limit the scope of this manuscript to only SNPs and short indels. The reason for this is two-fold: 1) Performing the necessary experiments for rigorously benchmarking KAGE on structural variants against competing methods requires extensive amounts of work and 2) genotyping small variants and structural variants is somewhat two different problems, with largely different sets of competing tools to compare against, which makes it natural to limit the scope of this manuscript to only one of these problems. We also believe that genotyping short variants -- although this limits the scope of this manuscript -- is still useful to a large community, and that genotyping structural variants will be a natural follow-up-step after this paper (a step we are currently working on).

When it comes to polyploid organisms, we think it should be possible to extend KAGE to work with not only diploid genomes, but this requires some work and we have chosen to not include this as a feature for now.

I find the manuscript simple and compelling. I commend the authors on focusing on an immediate and practical solution. However, it seems to me that more work needs to be done to make KAGE something that can be used in production. In particular, the question of what kind of variants can be used remains unresolved. Can we attempt to genotype structural variants? At what scale does the method break down? Due to the great interest in pangenomics in crop breeding research, there is presently great demand for a pangenomic genotyping method that works on polyploids. It would seem that the authors could use math for polyploid genotyping such as that described in freebayes or other methods. Handling either or both of these features would help to set KAGE apart from the competition not only in runtime and accuracy, but also in feature set. I also wonder if the authors have considered using extremely long kmers to genotype larger complex variants.

As mentioned in the previous comment, we have now extended KAGE so that it in theory should work with structural variants. We have however chosen to not include or mention this in the current manuscript, as we would like this manuscript to focus SNPs and short indels. We have made sure that the manuscript text and title specifically states that this manuscript and version of KAGE is for "SNPs and short indels" only.

It is a weak criticism, because it lies outside of what I think should be requested of the authors, but the use of python here seems problematic for the future maintenance and distribution of the software. A self-contained application in a lower-level language would potentially provide a much longer life to this software. In my experience, it can be extremely difficult to install old python packages. A system that can be built without reliance on external package managers might be advantageous in the long term.

We respect the reviewers' concerns, and agree that there are Python packages (especially old packages) that can be hard to install. Thus, we have tried to make the number of dependencies for KAGE small to avoid trouble when maintaining the tool in the future. Using Github actions, we have also performed extensive automated testing to make sure that KAGE and dependencies are possible to install and run on all common platforms with all recent Python versions (<https://github.com/ivargr/kage/actions>). We would argue that with the drastically increasing popularity of Python in the bioinformatics community during recent

years, Python is a good choice for new tools as it makes more people able to easily contribute to tool development.

Reviewer #2

The article presents KAGE, a speedy alternative to alignment-free genotyping SNPs and short indels using non-paired-end short reads, with performance comparable to previous kmer-based methods. In particular, the authors propose to use evidence at one other "helper" site to aid in genotyping a given site.

Overall, my enthusiasm is reduced by the vagueness of the manuscript (especially of the Methods) and the lack of comparison against imputation-based tools like GLIMPSE.

Major comments

The paper needs to clarify how "alignment free" methods are (or are not) related to avoiding reference bias. For instance, it is very possible for an alignment-free method to have reference bias. E.g. consider a k-mer-based genotyper that visits each polymorphic site in the genome and extracts two versions of the overlapping k-mers, one with the REF and one with ALT version of the site. If the flanking k-mer sequence consists of all REF alleles, then reference bias is still present. The paper should do more to explain that "alignment" is not the reason for reference bias; rather, the strict use of a single linear reference (whether we're doing alignment or genotyping) is the cause.

This is a good point, and the manuscript was a bit unclear on how kmers are selected. In fact, we make sure to select kmers so that reference bias is avoided by, for any variant, also including kmers that cover neighbouring variant alleles. The algorithm does not really care about what is the reference path in the graph, all paths are treated equally. I.e., if a variant A has another close variant B, there will be 4 kmers selected to represent the alleles of variant A (one kmer covering the variant allele of A and the variant alleles of B, one covering the reference allele of A and variant allele of B, one covering the reference alleles of both and one covering the variant alleles of A and reference alleles of B). In the modified version of the methods section, we have described this more in detail and also added a new figure that shows how kmers are selected when two variants are close.

The major methodological idea is the use of evidence at a "helper" site for genotyping. This takes a step outside the space of pure genotyping and toward something a little more like genotype imputation. Put another way: Li & Stephens based imputation tools (e.g. Beagle 5, GLIMPSE) are like KAGE but allow for a larger collection of nearby sites to influence the

genotype at a site, via the HMM. When we think of it this way, it becomes important to also understand how KAGE compares to tools that specialize in imputation based on sequencing data, like GLIMPSE.

The fact that KAGE internally does something similarly to imputation tools like GLIMPSE and Beagle is a good point that we should have mentioned in the manuscript. We thank you for pointing this out. There have recently been published papers that have used GLIMPSE or Beagle to show that low coverage sequencing, followed by genotyping, followed by imputation/genotype refinement leads to good accuracy (e.g. <https://doi.org/10.1016/j.ajhg.2021.03.012>), and we agree that it is important to show how KAGE relates to these pipelines.

From the literature search we have performed, it seems that there are two pipelines that do something similar to KAGE, i.e. using known population information to improve genotype calls; 1) a combination of Beagle version 4.1 and Beagle 5 (<https://doi.org/10.1016/j.ajhg.2021.03.012>) and 2) GLIMPSE. The Beagle-based pipeline is somewhat “hacky” since it relies on an old non-maintained version of Beagle to do genotype likelihood refinement. Also, Beagle does not really scale computationally to a reference population of thousands of individuals. Thus, we do not think the Beagle-based pipeline is a practical alternative to KAGE, and see it more as a proof-of-concept pipeline for low coverage sequencing.

From our understanding, the problems with this Beagle-based pipeline is some of the motivation for the tool GLIMPSE, which seems to be specifically built to improve genotype accuracy on genotypes produced by a genotyper (such as e.g. GATK or KAGE). We thus believe it would be interesting to run KAGE without the builtin “imputation” (i.e. without using any helper-variants to improve accuracy), and instead run GLIMPSE on the output from KAGE, and compare this to running KAGE normally. We thought it could be interesting to do a similar experiment for the other genotypers, but we have not been able to do this (PanGenie is heavily based on its builtin imputation, which cannot be turned off, and the other genotypers do not output variants with genotype 0/0, which is important for GLIMPSE to work).

We have now included a new part to the end of the results section where we run KAGE without the builtin imputation followed by GLIMPSE. We are really excited about these results, as GLIMPSE seems to perform very well on output from KAGE. Running GLIMPSE does add an hour or so to the whole process, but this setup seems like a good alternative to only running KAGE when runtime is not very critical.

We have also added a paragraph to the discussion section discussing these results.

The description of the KAGE methods is extremely short; only about a page or less of the Methods section is actually dedicated to KAGE. We're left to guess many details.

We appreciate the feedback. While we have tried to keep the methods section short since most of the principles behind KAGE are explained in the results section, we agree that this section is too short now. In the revised manuscript, we have added a new section to further

outline how KAGE works, and also included a figure that explains how kmers are selected from a genome graph.

The sense in which KAGE uses a "graph" was not well explained. The word "graph" is not mentioned in the Methods, except to say that a couple of the software components use graphs. Is KAGE graph-based only in the (rather trivial) sense that it considers both alleles at polymorphic sites? Where does the "graph" shape enter into the algorithm?

This is a good point, and we agree that information about how graphs actually play a role is lacking. KAGE uses a graph internally to represent variants, and the graph is also used when kmers are sampled from variants. Also, haplotype paths from the population are represented as paths through the graph. In the new section in the methods section, we now explain further how the graph is being used.

For speed comparisons, it seems that KAGE is being run on many cores at least some of the other tools are not. There should be at least one comparison where the tools use an equal number of cores. It's fine to also argue that KAGE scales well/better to many cores, but the speed comparison should be apples-to-apples in terms of what resources are actually made available to the algorithm. Also, the number of cores available to the algorithms should be laid out in the caption of any tables or figures presenting speed results.

We thank you for pointing this out, and agree that the manuscript is not clear enough on how many compute cores each tool is using. We have now modified the manuscript (under Figure 5) to clearly state that all tools except for Malva use the same amount of compute cores. We thus believe that except for Malva, we perform an apple-to-apples comparison when it comes to compute-time. When it comes to Malva, we have contacted the authors to ask about whether it can be run with multiple compute cores, and the answer was no (see <https://github.com/AlgoLab/malva/issues/8>).

More information on how the GIAB truth datasets are used to evaluate accuracy stats (data links, filtering procedures when plugging Hap.py, inclusion of high-confidence regions/bed files for different samples) would be helpful for quick reference.

We agree, and we have now included a short paragraph in the methods section to further explain how the GIAB truth dataset is being used: *Accuracy (precision and recall) was measured using Happy version 0.3.14 against GIAB truth datasets. Accuracy was measured within high-confidence regions using the Happy -f flag with the regions file that accompany each GIAB data set. We did not filter any variants on quality or other criteria before running Happy.*

Minor comments

"While PanGenie has decent accuracy" -- replace "decent" with something specific.

We have now changed this sentence to “While PanGenie has almost as good accuracy as KAGE, ...”.

PanGenie was sometimes misspelled.

Thanks for pointing out, this should be corrected now.

The text should express some idea of how time and memory usage break down across the different components of KAGE, like 'graph-kmer-index', 'obgraph', and 'kmer-mapping'. I am also curious how these scale with the number of alleles being considered in the "panel."

We agree, and we have now added more information about this in the results section:

KAGE is able to genotype a sample fast by pre-building all indexes (Methods). While creating these indexes is quite time-consuming (a couple of days), once indexes are built for a set of variants and a reference population, genotyping a new sample takes a short fixed amount of time, which is independent of the number of individuals in the reference population. For the experiment presented in Table \ref{table3}, KAGE spends about 2 minutes on genotyping and 10 minutes on counting kmers.

Reviewer #3

Grytten et al present a new method for alignment-free variant genotyping that performs as well as other methods but much faster and with lower memory. Overall the paper is well written and the results are compelling, but the specific use case is not well motivated. The authors do say this can run on a laptop "such as in a clinical setting", but I cannot imagine how this would happen given the indexing requirements. I do not mean to imply that KAGE has no use, I would just like the authors to better justify its need.

This is a good remark, and we have not been clear enough in the manuscript that KAGE can be run fast on a laptop *given* that you already have built the indexes. An important fact here is that all the indexes only need to be built once for a given set of variants we want to genotype. When you then want to genotype any new individual, you will use these indexes and do not need to rebuild them. Creating all the indexes is quite time consuming, and not possible to do on a laptop, but once they are built you can use KAGE to genotype any individual in a short time with low memory requirements. This is in many ways the same principle as for variant calling with e.g. GATK, where indexing the reference genome for read mapping is only done once. On the Github repository, we provide indexes for 1000 Genome variants, and plan to upload different sets of indexes in the future (with varying number of variants, and maybe some indexes for subpopulations). We have now changed the sentence you refer to in the manuscript to clarify: “This means that with KAGE, given pre-built indexes, it is now possible to genotype a sample quickly and easily on a standard laptop, such as in a clinical setting.”

I do not understand the 1KG reference ([1]) in the background. The authors are discussing variant detection and give a 90% figure. The closest thing I can find in [1] is an imputation result, which is not variant detection. The authors could certainly derive this number in their experiments, and I suggest they do those experiments to give readers a more complete picture. Regardless of the validity of the reference, the 90% cannot be given without context. What allele frequency range is this 90% valid for and how does the % change for common and rare variants? The % is almost certainly ancestry specific. Genotyping individuals from populations with poor sampling will perform worse. How much worse is the recovery for different populations? What regions of the genome are considered in this %? Several papers over the previous few years have many many mega bases of new sequences in non-European samples. I suggest the authors incorporate these complexities (allele freq, ancestry, genomic region) into their larger analysis. Ancestry in particular must be considered. The negative effect of poor sampling for marginalized populations is well known in population genetics. We must work to improve sampling (which is clearly out of scope here) while also quantifying the effect on all methods going forward.

You are correct that the 1KG reference [1] does not back up the claim we make in the Background section. We think we at some point came to a 90% figure based on the numbers given under the section “A typical genome” in the 1KG paper, but when reading this again now, it is not clear how any specific number can easily be concluded on, as you point out. The point we are trying to make is simply that since a lot of human variation has been characterised by the 1000 Genomes Project, an individual with variants well represented in the 1000 Genome Project can typically be genotyped with good recall (e.g. most variants can be genotyped) using 1000 Genomes variants, something that we have observed when running genotypers like Malva or Graphtyper where the recall on GIAB’s high-confidence regions (where GIAB claims to have detected “all” variation) is typically well over 90%. However, this number will of course vary between individuals, and we think throwing out a number is not what is important here, but rather indicating that genotyping can make sense to do against a reference population like 1000 Genomes. We have now changed the sentence in the background section by removing the 90% figure and the the 1000 Genomes reference, and instead citing Malva and Graphtyper which both show what we are claiming (that a “a large amount of an individual’s genetic variation can be detected [by genotyping against an existing reference population]”).

Regarding your other comment about the complexities related to ancestry, allele frequency and genomic regions, we agree. The accuracy achieved when genotyping against a reference population will vary depending on how well the individual is “represented” by the reference population, and is likely to vary depending on ancestry. It would be very interesting to perform a larger benchmark study where various reference populations (e.g. with different ancestry) are used, and see how accuracy varies depending on the reference population. However, we believe this is partly outside the scope of this article, as what we try to show here is instead that given a reference population like the 1000 Genomes, KAGE performs well compared to other genotypers that genotype the same individual against the same reference. Both KAGE and other genotypers are likely to perform differently on different individuals and reference populations, but there is no reason to believe that the difference in accuracy between e.g. PanGenie and KAGE would change considerably when the reference

population changes. We have added a discussion of this in the Discussion section:

“Second, KAGE relies on a relatively good database of known variation from a population. This is important to remember, as not all individuals are well represented in e.g. the 1000 Genomes Project, and genotyping accuracy for such individuals is likely to be lower. The fact that more and more genotypers are using population information to improve accuracy highlights the importance of good reference populations (pangenomes) that represent and cover all ancestries. In the same way, KAGE ...”

The authors hypothesize that performance is based on kmer uniqueness. I suggest they test this hypothesis. They can, for example, compare their naive and KAGE methods and show that outliers are correlated with repeats.

We very much appreciate this suggestion! We have now performed an additional experiment where we measure the performance on two sets of variants: Variants that can be represented by unique-kmers and variants that are not covered by any unique kmers (meaning they only have kmers that also exist someplace else in the pangenome, and thus are more difficult to genotype). We measure the performance on these two types of variants for the “Naive KAGE” method, the version of KAGE that models expected kmer counts but that do not use helper variants to improve performance, and the full KAGE method. We show the results in a new table (Table 1).

In another section the authors speculate about why KAGE and PanGenie do well but do not test this speculation. This speculation should be tested. They can, for example, start with a small amount of population information (eg just a few haplotypes for PanGenie) then scale up to the current result. This would not be too computationally expensive but would provide evidence in support of the speculation.

We think this is a very good suggestion, but we believe we already do this in Figure 3, where we run both KAGE and PanGenie using a varying amount of individuals in the model. What we see in this figure is that both KAGE and PanGenie perform better as more individuals are included. We were not able to run PanGenie on more than 100 individuals, but based on the trend in this figure, we would guess that PanGenie could perform better if it was able to scale to use more individuals in the model.

Was GATK used to define the truthset? If so that would explain GATK's performance and should be pointed out to the readers.

That is a good question. From what we can find, both GATK and FreeBayes were used on different sequencing technologies to create a curated high-quality data set. It is hard to say whether the involvement of GATK would make GATK perform noticeably better, and if so to what degree. We don't think this potential bias is a big problem here, since we are not primarily concerned with comparing KAGE to GATK (but rather the other alignment-free genotypers), but think it is a good idea to mention this in the paper so that the reader is aware that there is a possibility that the GATK results can be biased. We have added this sentence to the discussion section: *“ It should be noted that GATK was used among other*

tools to create the GIAB truth dataset, which could potentially influence the measured performance of GATK in our experiments.”

I was not able to test this software because "kage test" gave an error.

```
~$ kage test
```

```
Traceback (most recent call last):
```

```
[...]
```

```
OSError: [Errno 63] File name too long: '1288721527800148__frequencies_squared'
```

This is a strange error that we haven't experienced before. The Python exception suggests that a file name is too long, but the file name used internally here is not really very long, and should be accepted by all major operating systems. We have tried reproducing the error on several systems (Windows, macOS and Ubuntu) for various Python versions (3.8, 3.9 and 3.10) using Github actions (<https://github.com/ivargr/kage/actions/runs/2535015968>), but are not able to reproduce the error. Some googling suggest that it might not be the length of the filename that is the cause of this exception, but other problems with files (permissions, etc). We must admit we really don't know what the cause could be here, and cannot provide a direct fix, but we are happy to continue to try to solve this issue if you are still running into it.

Second round of review

Reviewer 2

This us a very thorough revision, which considered and acted on all of the input. I was particularly heartened to see the more elaborated methods section as well as the new method that uses GLIMPSE. Very nice work.

Reviewer 3

All of my issues have been addressed.